# A Comprehensive Review of the Latest Trends in Spray Freeze Drying and Comparative Insights with Conventional Technologies

**DOI:** 10.3390/pharmaceutics16121533

**Published:** 2024-11-29

**Authors:** Maria Ioannou Sartzi, Dimitrios Drettas, Marina Stramarkou, Magdalini Krokida

**Affiliations:** Laboratory of Process Analysis and Design, School of Chemical Engineering, National Technical University of Athens, 9 Iroon Polytechneiou St. Zografou Campus, 15780 Athens, Greece; ioannssm@gmail.com (M.I.S.); dimitrisdrettas@gmail.com (D.D.); mkrok@chemeng.ntua.gr (M.K.)

**Keywords:** atomization, freezing, sublimation, encapsulation, pharmaceuticals, spray drying, freeze drying

## Abstract

Spray freeze drying (SFD) represents an emerging drying technique designed to produce a wide range of pharmaceuticals, foods, and active components with high quality and enhanced stability due to their unique structural characteristics. This method combines the advantages of the well-established techniques of freeze drying (FD) and spray drying (SD) while overcoming their challenges related to high process temperatures and durations. This is why SFD has experienced steady growth in recent years regarding not only the research interest, which is reflected by the increasing number of literature articles, but most importantly, the expanded market adoption, particularly in the pharmaceutical sector. Despite its potential, the high initial investment costs and complex operational requirements may hinder its growth. This paper provides a comprehensive review of the SFD technology, highlighting its advantages over conventional drying techniques and presenting its latest applications focused on pharmaceuticals. It also offers a thorough examination of the principles and the various parameters influencing the process for a better understanding and optimization of the process according to the needs of the final product. Finally, the current limitations of SFD are discussed, and future directions for addressing the economic and technical barriers are provided so that SFD can be widely industrialized, unlocking its full potential for diverse applications.

## 1. Introduction

Spray freeze drying (SFD) is a technique that produces powders consisting of highly porous, microscale particles. SFD bridges the benefits of traditional spray (SD) and freeze (FD) drying methods, combining rapid freezing with controlled sublimation [1]. The use of low temperatures, along with the ability to encapsulate compounds, establish SFD as an ideal method for pharmaceuticals, food, and specialized industrial applications, where sensitive substances are often found [2]. The morphology of the produced particles improves vital characteristics such as aerodynamic properties and dissolution rate, making them suitable for methods of delivery where size and shape are of great importance, including the nasal and intradermal routes [3,4].

Selecting the appropriate drying process can be a challenge when dealing with thermosensitive compounds. SFD is not only appropriate for such substances, but also, as a gentler technique, it minimizes processing losses, maintains the product integrity, and adds distinct product qualities (enhanced hydration, porous structure, rapid reconstitution, etc.) [2]. These characteristics offer this technique a competitive edge over conventional drying methods, making it suitable for the processing of various drugs, water soluble or not, the production of powders with specific characteristics, and the encapsulation of bioactive compounds in drugs and functional foods [5].

SFD has gained attention for its ability to enhance drug bioavailability, stability, and delivery efficiency, particularly for poorly soluble compounds. Key studies highlight its effectiveness in achieving high fine particle fractions, rapid dissolution, and superior aerosol performance, as demonstrated in formulations for cystic fibrosis treatment [6,7] and inhalable antifungals [8]. Additionally, vaccines produced via SFD have shown remarkable results in immune responses in the respiratory tract, making this method appealing in the preparation of immunotherapy treatments [9,10]. SFD products’ high porosity and amorphous particle state have also improved bioavailability in small-molecule drugs [11]. These findings demonstrate the potential of this method to enhance therapeutic effectiveness, stability, and targeted delivery across a range of applications.

In recent years, SFD has attracted significant attention from both the industrial and research community. This is proven by the fact that the global “SFD Equipment Market” achieved a valuation of USD 20 billion in 2023 and is projected to reach USD 38.71 billion by 2031 with a compound annual growth rate (CAGR) of 9.89% for this period. North America and Europe are leading in the market adoption of SFD equipment due to strong research and development activities and the presence of key industry players [12]. Furthermore, the rising number of publications during the last 20 years clearly demonstrates the upward scientific trend in SFD, starting from 2003 with a steady increase, which accelerated around 2017 and continued sharply for the last 7 years until 2023. The advancements in SFD in recent years were mainly driven by the global trends in health and pharmaceutical fields, as shown in Figure 1b, where pharmaceutical research represents the primary focus, accounting for 31% of total references, according to Google Scholar and Scopus metrics.

However, despite its advantages, SFD faces significant economic and technical barriers, including high initial investment costs and complex operational requirements, that need to be addressed for widespread industrial adoption [1,13,14,15]. In order to unlock the full potential of SFD, the mechanisms of this process, as well as the operational parameters that crucially affect its efficiency, should be well understood so that advancements can be proposed for the optimization of SFD technology.

This paper provides a comprehensive review of SFD, discussing its principles, process parameters, and diverse configurations and presenting its main applications across pharmaceuticals, foods, and other sectors. Furthermore, a comparison of SFD with SD and FD is performed in order to demonstrate its strengths over conventional techniques. Finally, the key factors that influence its performance, as well as the future needs, are discussed, providing ways for optimizing the method to meet specific product requirements. The study of SFD is of great importance thanks to its potential for developing high-quality and stable powders of bioactive components widely demanded in the modern pharmaceutical and nutraceutical industries.

## 2. Principles of SFD

The process of SFD can be better understood as the sum of three sub-processes: (i) atomization, (ii) freezing, and (iii) drying (sublimation), as presented in Figure 2.

During atomization, the substance of interest is dissolved in a solvent, and the solution is atomized into a chamber, where it comes into contact with a cold fluid. The large temperature gradient causes the droplet to be rapidly cooled until it reaches the ice nucleation temperature and then to be supercooled [16]. This leads to rapid crystal growth, and, as a result, the droplet is quickly solidified, retaining its initial shape. The beginning of the solidification process requires the formation of a crystal nucleus. The existence of solvent-gas vapor fog throughout the chamber, in combination with the gas flow fluctuations, favors triggering nucleation on the droplet surface. Therefore, the solidification process starts from the outer shell, forming an ice front, and the freezing process progresses inwards [17]. As the solid phase is formed, the solute diffuses to the liquid phase until the maximum freeze concentration is reached, and afterward, the droplet exists in a glassy phase [18]. The rapid freezing of the particle limits the phase separation between the carrier and the substance of interest [19]. The complete solidification of the droplet below the glass transition temperature (Tg) before it reaches the container at the bottom of the chamber is crucial; otherwise, the risk of particle agglomeration and adherence to the container wall increases dramatically, compromising the quality of the final product. Particle agglomeration also occurs during the free-falling phase of the droplet due to inter-particle collisions. Deflector jets inside the chamber can be used in order to assist with minimizing droplet-droplet and droplet-wall collisions [17,20].

After this process, lyophilization of the particles occurs, which is divided into primary and secondary drying phases. During primary drying, the frozen solvent is removed through sublimation, which typically occurs below the triple point [21]. However, the presence of a vacuum is not an absolute requirement as a sufficiently low partial pressure at the site of the solvent vapor formation in the drying medium causes the mass transfer driving force required to remove the solid solvent through vapor. Therefore, FD can be performed at atmospheric pressure [22]. The sublimation of the solvent cools the particle, as the energy of sublimation is required for vaporization, reducing the partial pressure gradient and the drying rate. Thus, heat must be supplied to the sample; however, a fine balance has to be kept as the temperature must stay below Tg in order to avoid particle collapse. The sublimation interface progresses deeper into the particle, decreasing the sublimation rate as increased resistance to vapor migration occurs. The solute retains its spatial positioning after the solvent sublimation preserving the particle shape [23]. The advantage of SFD is that lower cake resistance exists due to the small particle size and the higher porosity [17]. After the majority of moisture is sublimated, the secondary drying follows, during which the remaining solvent bound in the glass phase is desorbed. Secondary drying is much less efficient as increased resistance to mass transfer is observed. The possibility of structural collapse is lower as the particle is mechanically stable, and thus, the temperature can be increased; however, it still has to be kept below Tg [23]. An optional step in SFD is annealing. During annealing, the temperature is maintained between Tg of the glass phase and the melting point of the solvent. This step promotes the growth of larger crystals, further concentrating the glass phase, which in turn raises Tg and permits higher drying temperatures [24]. The larger pores formed after the sublimation of the solvent decrease the mass transfer resistance, and therefore, faster sublimation rates are achieved [1,25].

## 3. Set Ups of SFD

As a three-step process that includes atomization, freezing, and drying, SFD permits adjustments at any stage to meet the requirements of the final product. In the following paragraphs, the various configurations of atomization, freezing, and drying are discussed.

### 3.1. Atomization Nozzle Types

The atomization step is vital since it determines the final shape of the particle, making its optimization essential. Different nozzle types have been tested in order to achieve the desirable droplet characteristics and size dispersity, with each nozzle type having specific advantages and disadvantages that will be discussed in Section 4. The most commonly used atomizer types are pneumatic (or multi-fluid) (Figure 3a) and hydraulic (Figure 3b). Ultrasonic nozzles and droplet generators have also been reported in the literature.

#### 3.1.1. Pneumatic or Multi-Fluid Atomizer

Pneumatic nozzles operate with low-pressure liquid feed and use the energy of a high-speed gas stream, which comes into contact with the liquid, resulting in droplet formation. These nozzles can accommodate a variety of liquid feed flows and can be used in both laboratory-scale and large commercial dryers. They can also produce droplets of a desired size, with a micron-level precision control and a small size distribution [26]. Additionally, three- or four-fluid nozzles enable the atomization of two different feed streams [5].

#### 3.1.2. Hydraulic Atomizer

The application of hydraulic pressure on the liquid is used in order to atomize the solution, hence avoiding the need for an auxiliary gas stream. Droplets are generated by a pressure swirl after being introduced to a swirl chamber within the nozzle. In comparison with the twin fluid nozzle, which requires a feed pressure of 0.5 bar, this atomizer type operates at a liquid feed pressure of 40–200 bar. The high pressures result in high mechanical stresses on the feedstock [26].

#### 3.1.3. Ultrasonic Atomizer

Ultrasonic atomizers utilize high-frequency electrical signals to create mechanical vibrations. These vibrations atomize the liquid feed and generate droplets of a variety of sizes, depending on the frequency and feed flow. Ultrasonic nozzles usually consist of piezoelectric transducers placed between electrodes [5].

#### 3.1.4. Droplet Steam Generator

The generation of nearly monodisperse droplets can be achieved with inkjet printing and automated analytical reagent dispensing systems. These methods can produce droplets with a diameter of a few micrometers, depending on the surface tension and the adhesion of the liquid to the nozzle wall [27]. Generators used in SFD include those that create droplets by applying heat to vaporize the solution (thermal inkjet) and those that micro-fluidic-aerosol-nozzle [28].

### 3.2. Freezing Medium Phase

The freezing medium must always maintain a temperature low enough to cause practically instantaneous droplet solidification. However, the interface between the droplet surface and the freezing medium can affect the physical structure of the substances that are to be frozen, such as proteins. The medium rheology also influences particle agglomeration. Based on the freezing medium phase, the process can be subcategorized as (a) spray freezing into vapor, (b) spray freezing into vapor over liquid, and (c) spray freezing into liquid, as demonstrated in Figure 4 [5].

### 3.3. Drying Process

The versatility of the SFD process allows for modifications in the final drying step, where besides the choice of the drying cycle, the powder form of the droplets enables drying to be carried out either in a packed bed or a fluidized bed [29]. Although this typically requires moving the droplets into a separate compartment for lyophilization, there have been attempts to incorporate this step into a fully continuous process, as shown in Figure 5 [5]. The addition of a heated vibrating bed can allow for the movement of the frozen in a separate compartment, where they can be dried, as seen in Figure 5a. Developing this concept further (Rey’s concept), a secondary drying area can be added, and the final product filled in a moving belt of vials, as seen in Figure 5c. Another approach, currently utilized on a FD industrial scale, uses a rotary drum [5].

## 4. Parameters Influencing the Yield and the Characteristics of the Final Product

SFD provides numerous adjustable parameters, allowing for fine-tuning of both the final morphology and the time needed to achieve the desired features of the developed particles. As previously mentioned, SFD can be divided into three steps: atomization, freezing, and drying. While the freezing and drying stages of the process have a minimal effect on particle structure, the critical parameters affecting the particle form are primarily associated with the atomization step.

Additionally, the particle size and shape may also be affected by the use of additives for encapsulation and cryoprotectants, though their influence depends on a plethora of factors such as the core-wall interactions, the hydrophobicity, and the synergistic effects between the used substance and, as a result, they should be examined on a case-by-case basis. Nevertheless, several general principles for controlling their features have been reported and discussed in this section, individually for each process step.

In Table 1, the most important parameters and their influence on the basic characteristics of the final SFD products are presented.

### 4.1. Atomization

#### 4.1.1. Solution to Be Sprayed

The parameters that can be controlled in the feed solution include solid content concentration, solution viscosity, and choice of solvent. Starting with the solid content concentration, an increase in the value of this parameter leads to higher sphericity, density, and particle size while simultaneously decreasing porosity and residual moisture [8,11,30,31,32].

Regarding viscosity, an increase in the solution viscosity results in larger particle size, less inner particle void after sublimation, and a more structurally stable droplet that can retain its initial spherical shape. A feed solution with increased viscosity also requires more energy for atomization, resulting in a longer time for the particles to grow in the nozzle before being propelled and frozen into the chamber [6,11,34].

Lastly, the choice of solvent plays an essential role in the particle size since an organic solvent, instead of water, generally leads to smaller particle sizes [39]. This differing behavior is caused by the solvent volume change during the phase transition from liquid to solid, with water expanding while most organic solvents shrink.

#### 4.1.2. Nozzle Configuration

Different types of nozzles have been employed in SFD to achieve the desired results, including (i) hydraulic nozzles, (ii) two-fluid nozzles, and (iii) ultrasonic nozzles. In general, smaller nozzle diameters create smaller particles, as smaller droplets are formed at the nozzle edge [30].

For hydraulic nozzles, droplet size decreases by increasing the liquid feed rate [33]. This nozzle type has the disadvantage that liquid feed is the only customizable parameter, restricting optimization options. Furthermore, the droplet size distribution tends to have low homogeneity [5].

In two-fluid nozzles, the critical process parameters are the atomizing gas flow rate and the liquid flow rate. Increasing the gas flow rate leads to greater shear stress applied to the particle, which in turn causes premature atomization [34]. This results in particles of a decreased size and higher fine particle fraction. Conversely, increasing the liquid flow rate creates larger particles, as more liquid at the nozzle creates particles of higher diameter [36]. These opposing effects also impact the final yield, as the smaller, more friable particles may collapse, leading to decreased yields [35].

A special type of nozzle is the four-fluid nozzle, which has the benefit of allowing different solvents to be used at the same time, thus overcoming the challenge of finding a common solvent for poorly soluble substances and their additives [39]. This nozzle configuration has also been found to produce smaller particles, likely due to the increased probability of particle collisions before solidification [37].

Another type of nozzle is the ultrasonic nozzle. It achieves the desired droplet characteristics through vibration, with the critical parameter being the frequency of vibration. Lower frequency causes an increase in the droplet size as the standing sinusoidal longitudinal wave responsible for the atomization has less energy. The droplet size distribution is narrower compared to the two-fluid nozzle due to better control over the droplet formation. The two-fluid nozzle requires a high atomizing gas flow rate, and thus, small sample sizes cannot be handled as easily [4]. The effect of the amplitude of nozzle vibration is not easily predictable. Higher amplitudes theoretically lead to smaller particle sizes. However, it is reported that at higher amplitudes, the solution might fail to atomize because of slippage, leading to increased particle size. It is also worth noting that ultrasonic nozzles are less effective at higher liquid feed rates, which can slow down the process [40].

Another type of atomizer that has been experimented with in SFD is the thermal inkjet (TIJ). TIJ offers the advantage of atomizing smaller quantities of solution with fine control over droplet size [41]. It has also been found that increasing the distance between the TIJ and the liquid nitrogen reduces the particle size [42].

### 4.2. Freezing

While the freezing step has minimal impact on particle morphology, it is essential that it occurs nearly instantaneously to maintain the droplets’ initial shape after atomization. Among spray freezing into vapor, spray freezing into vapor over liquid, and spray freezing into liquid, the most commonly used technique for SFD is spray freezing into vapor over liquid. However, in a comparison study, it was found that spray freezing into liquid minimized protein aggregation and enzyme activity loss due to a significantly decreased air-solvent interface. It also led to particles with a slightly decreased porosity [43]. On the other hand, spray freezing into liquid requires a special insulated nozzle to avoid the risk of the solution freezing inside the nozzle and causing damage to the apparatus.

The droplets, which are not yet fully solidified, face the risk of collisions and agglomeration. These are unfavorable phenomena as they result in broken up or conjoined particles. The employment of a jet vortex inside has been reported to minimize particle collisions and lead to more uniform particles and a narrower size distribution [44].

### 4.3. Drying

Drying conditions have a limited effect on the particle structure; however, the drying step accounts for the majority of processing time. The drying step is carried out either in a fluidized or a packed bed. Most commonly, the solidified particles are collected and placed in a lyophilizer, where they are dried in a packed bed.

In general, increasing the primary drying temperature decreases the drying time due to faster sublimation rates [38]. However, it also exacerbates shrinkage as the particles approach Tg. At temperatures higher than Tg, the viscosity drops substantially, leading to the deformation of the glass matrix and the collapse of the porous structure [4].

The incorporation of an annealing step before drying can further lower the drying time [25]. Annealing may have a substance-specific effect on particle size, with darbepoetin alfa particles increasing five-fold in size [45], while trypsinogen particles shrank [25].

The existence of interparticle voids in the packed bed forms an unsaturated porous media as opposed to traditional FD, which involves drying a non-saturated, nonporous structure [46]. These interparticle voids are considered the major rate-limiting factor in packed bed drying [47]. A stratified packed bed, where the larger particles are placed on the upper surface, has been theoretically shown to reduce drying times [48]. The use of inert particles has also been found to assist in drying. By filling the interparticle voids, inert particles improve thermal conductivity, leading to faster heat transfer and increased sublimation rates [49]. In another study, radio frequency-assisted ultrasonic FD was employed and significantly reduced primary drying time without compromising protein stability [50].

For fluidized bed drying, a critical parameter regarding drying time is the gas flow rate, as a larger quantity of air allows for more ice to be sublimated [22]. Decreasing the pressure results in higher drying speeds, as the lower partial pressure of the water vapor in the drying air creates an increased mass transfer force. However, secondary drying is not possible, as the electrostatic forces cause particle adhesion to the walls [29].

## 5. Applications of SFD

SFD is mainly employed for the development of pharmaceuticals, foods, and other products, such as ceramics, catalysts, and insulating materials, frequently involving the encapsulation of various valuable bioactive compounds. During encapsulation, the encased substance, called the core, is entrapped inside a carrier, known as the wall/shell material, creating encapsulates of various sizes and properties [51].

The most significant and recent applications of SFD, along with the most widely used materials, are analytically presented in the next Sections, which are divided into the three main application categories of SFD, namely pharmaceutical, food, and other material applications.

### 5.1. Pharmaceutical Applications

In the pharmaceutical field, SFD can produce powders for unconventional methods of delivery, including inhalable, intranasal, and intradermal delivery, while also allowing for improved dissolution characteristics. The pharmaceutical applications mainly refer to asthma and other lung disease treatments, as well as the development of vaccine powders and antibiotics. The used core materials include volatile compounds, aiming to produce inhalable drug powders for various conditions, such as respiratory tract infections, lung diseases, metastatic lung tumors, or vaccines [10,52,53,54,55]. These materials can be categorized into the following groups:

(i) antibiotics and antimicrobials: metronidazole [56], ciprofloxacin [57], clarithromycin [55], etc.

(ii) antiviral agents and vaccines: yellow fever virus (vYF) [58], hepatitis B surface antigen [54], Monovalent influenza [9,10,31]

(iii) respiratory drugs: mometasone furoate (MF) [59], montelukast sodium (MTK) [60], etc.

(iv) anti-inflammatory and pain management drugs: celecoxib [11]

(v) proteins and enzymes: bovine serum albumin (BSA) [50], Lupinus mutabilis proteins [61], trypsinogen [25], trypsin [28]

(vi) natural compounds: resveratrol [62], oleanolic acid [63] etc.

(vii) si-RNA [35,53,64].

Regarding the wall materials of SFD, different wall materials have been utilized according to the final use of the product. Inulin, mannitol, and other sugars are commonly used, functioning as lyoprotectants and stabilizing the particles during the lyophilization process [10,18,35,54,55,64,65]. In addition, Poly-lactic-co-glycolic acid (PLGA), polyvinyl alcohol (PVA), polyethyleneimine (PEI) and chitosan are widely used polymers and/or polymer-based materials in nano and/or micro-particle formulations for pulmonary delivery thanks to their stability, which promotes the controlled release and the high drug loading capacity [6,62,64,66]. The specific pharmaceutical applications of SFD are analytically presented below.

#### 5.1.1. Asthma Treatment

Different active ingredients have been used in the SFD method to produce powders for inhalation delivery to treat asthma and its symptoms. These powders have been shown to possess superior aerodynamic characteristics as opposed to powders produced by conventional methods, such as SD or FD [27]. Aziz et al. (2023) atomized montelukast sodium using raffinose as a carrier and reached a fine particle fraction of 60% [60]. Sharma et al. (2013) produced terbutaline sulfate powder for inhalation delivery to treat asthma symptoms, achieving a fine particle fraction of 22.9% [42]. Emami et al. (2019) produced an adalimumab formulation using various amino acids. These formulations with leucine, phenylalanine, or glycine in the presence of trehalose showed good stability after storage at 40 °C and 75% relative humidity [67].

#### 5.1.2. Other Lung Diseases

Due to their beneficial characteristics, inhalable SFD powders can be used in treating other lung-related illnesses. Several formulations developed via SFD demonstrate superior aerosol performance and dissolution rates, as shown by Yu et al. (2021) and Shahin et al. (2021), who achieved high fine particle fractions and significant dissolution improvements in dual-drug powders for cystic fibrosis treatment. Studies have been conducted on voriconazole, an active ingredient used to treat mycosis, including pulmonary aspergillosis. After optimizing the parameters, an FPF of over 40% was achieved. Liao et al. (2019) reported enhanced lung deposition and extended pulmonary retention in the SFD-produced voriconazole powder, suggesting clinical potential for inhalable antifungal therapies [8].

Yu, Tran et al. (2016) produced a stable curcumin chitosan complex, generally used to treat pulmonary disorder symptoms, which had an increased dissolution ability [32]. Yu, Teo et al. (2016) produced a ciprofloxacin nanoplex powder used for bronchiectasis treatment. Superior aerodynamic characteristics were achieved with a FPF of up to 29% [68]. Rahmati et al. produced inhalable salmeterol xinafoate powder with a significantly improved dissolution rate, reaching up to 90% drug release in 30 min [69]. Finally, Okuda et al. (2018) prepared siRNA powders with SFD that successfully silenced target genes in lung tumors, slowing tumor growth, that did not cause major adverse effects, such as inflammation or lung damage, which is often observed with inhalable treatments [64].

#### 5.1.3. Vaccine Powders

Vaccine powders have also been prepared using the SFD method. Clenet et al. created a proof of concept for a Vero-cell yellow fever vaccine. A live-attenuated virus was formulated into stable micropellets [58]. Patil et al. produced a whole inactivated virus influenza vaccine for inhalation delivery and examined the various adjuvants. Immunization on mice was achieved without the triggering of an overt inflammatory response [70]. Vaccine formulations created using spray freeze drying by Saluja et al. (2010) and Amorij et al. (2007) elicited robust immune responses, significantly boosting mucosal IgA and systemic IgG levels. These results suggest that SFD vaccines could offer improved cross-protection and strengthened immunity in the respiratory tract.

#### 5.1.4. Antibiotics

SFD has also been used to improve the dissolution kinetics and bioavailability of poorly soluble drugs, including antibiotics. Lucas et al. produced celecoxib powder with improved drug dissolution and in-vivo drug absorption characteristics due to the high porosity and amorphous state induced by SFD [11]. Kozak et al. improved rivaroxaban dissolution rate and achieved better bioavailability in rats [71]. Adeli et al. formulated azithromycin powder, an antibiotic for various diseases, to improve its bioavailability and dissolution characteristics [72].

#### 5.1.5. Other Pharmaceuticals

The versatility of the SFD method can be seen in its wide-ranging applications. Metronidazole powder was used to produce ointment for the treatment of rosacea. The produced ointment achieved edema reduction in significantly lower dosages than commercially available creams [56]. Keyhan-Shokouh et al. (2021) used Rizatriptan benzoate, an active substance used for migraine treatment, to create powder for pulmonary delivery. Using trehalose and phenylalanine as additives a FPF of 61.1% was achieved [73]. Poursina et al. (2017) produced parathyroid hormone inhalable powder used for osteoporosis treatment. The systemic delivery on rats was evaluated, and an absolute bioavailability of up to 47.25% was achieved [74]. Intiquilla et al. (2022) created SFD chitosan nanoparticles for colonic delivery of peptides, achieving controlled release and antioxidant retention with minimal cytotoxicity [61].

Finally, other delivery methods have been examined. Di et al. (2021) produced resveratrol powder for intranasal delivery, with significantly improved antioxidant activity and dissolubility improved up to 1800 times [62,74]. Straller et al. (2017) produced powder for ballistic delivery using bovine serum albumin and carbonic anhydrase as model proteins for future research [75].

All the pharmaceutical applications of SFD that have been reported in the literature are summarized in Table 2.

### 5.2. Food Applications

The method of SFD is also employed in the development of foods and supplements since fine powders are obtained while their thermosensitive valuable ingredients are protected. This is mainly performed through encapsulation, which preserves organoleptic characteristics, enhances the bioavailability of poorly soluble bioactive substances, and produces functional foods [82,83]. In general, the core materials mainly used during SFD are (i) probiotics, such as Lactobacillus casei [84,85] and Lactobacillus plantarum [86,87], (ii) bioactive compounds, such as carotenoids [82], vitamins (vitamin E) [88], and omega-3 fatty acids (docosahexaenoic acid, DHA) [83], phenols (vanillin) [13], and (iii) enzymes, such as Saccharomyces cerevisiae [89] and transglutaminase [14]. The principal purpose of the employment of SFD in the before-mentioned studies is to improve the stability of the encapsulated ingredients for the production of food ingredients, foods, and nutritional supplements of superior quality.

In addition, other studies investigated micellar casein, a common nutritional supplement ingredient with poor rehydration ability. The produced SFD powder achieved a dissolution of 80% in 15 min [90]. Similarly, a study on milk powders found that a 15% higher re-dispersibility, as opposed to conventional freeze-dried products, could be ascertained [91]. Another study produced instant coffee powder with superior foam stability, a characteristic considered preferred by customers [92]. The abovementioned works are shown in the following Table.

Regarding the wall materials, whey protein is frequently used as it offers the advantage of excellent emulsification properties and film formation [13,88]. Other wall materials include polysaccharides, starches, cellulose, and gums [88].

The core materials used in SFD for the development of different functional foods and food ingredients, as well as the target applications of the final SFD products, are found in Table 3.

### 5.3. Other Applications

Except for pharmaceuticals and foods, the physical characteristics of SFD powders find use in material applications, such as ceramics, catalysts, and insulation materials. More specifically, SFD has been employed for encapsulating TiO_2_ for producing composite material powder or porous micro-granules, with a coating of PFSA (perfluorosulfonic acid) polymer and silica, respectively [93,94]. In addition, Luo et al. produced catalysts for oxygen reduction reaction using Pt-based nanoparticles immobilized on reduced graphene oxide. The reaction kinetics achieved are promising for utilization on fuel cell technologies [95]. Multiple studies have examined the application of MgAl_2_O_4_ spinel to create transparent ceramic materials with favorable mechanical properties and high-temperature resistance [96,97]. Other uses of SFD involve cellulose-based aerogels [98] and silica aerogels with excellent thermal stability and low thermal conductivity for insulation applications [99], SiC ceramics with improved mechanical properties [100], and high energy density materials [101] (Table 4).

## 6. Comparison of SFD with Conventional Techniques

### 6.1. Comparison with SD

SD is a one-step, continuous drying process where a liquid or thin slurry feed is atomized into a hot chamber containing a gaseous drying medium. During atomization, the liquid jet is converted into tiny droplets, resulting in rapid evaporation of moisture upon contact with the heated air [1,2,26]. This contact between droplets and the surrounding air causes significant heat and mass transfer, with high interfacial surface area leading to high evaporation rates. Consequently, the process has a short residence time in the heating chamber (5–30 s), effectively preserving organoleptic properties [26]. The feed can be in the form of paste, solution, or suspension, with the resulting product taking the form of dust, aggregate, or grain [2].

SD has been widely used to transform a variety of pumpable liquids into free-flowing powders and to produce nanocomposite microcarriers with aerodynamic diameters appropriate for pulmonary deposition [102]. The method supports a variety of coating materials and enables control over particle size and flowability, bulk density, degree of crystallinity, and residual solvent or moisture content of the final product while also enhancing the solubility and dissolution rate of poorly soluble drugs [15,26,82,83]. In addition, it offers process flexibility and complete operation automation while also being inexpensive and easy to industrialize [14,83]. However, the method may result in significant losses of volatile substances and thermal degradation of thermolabile compounds [1].

Atomization and high drying temperatures in SD can cause the degradation of heat-sensitive materials [1,2,14,103]. According to different studies, high temperatures result in the oxidation of polyunsaturated fatty acids (PUFAs), about 10% loss of carotenoids, and a decrease in cell viability [82,83,87]. The SD process exposes the materials to shear, interfacial, and thermal stresses. Moreover, even though rapid evaporation shortens the drying time and lessens the loss of volatile compounds, the solute or nanoparticle enrichment occurs at the droplet’s surface, and the particle buckles or crumples upon drying. The significant difference in diffusion coefficients between molecularly dissolved matrix materials and dispersed nanoparticles induces component segregation in the final dried particle, which may contribute to irreversible particle aggregation [79].

SFD has been proven to be effective for a multitude of applications, as seen in Table 2 and Table 3. In a study on docosahexaenoic acid encapsulation, it was found that the resulting product had comparable residual moisture levels to those of other conventional drying methods. Specifically, SFD powder had 3.66% FD 5.49% and SD 2.40%. The poorer result, in comparison to SD, is suspected to be due to the low temperatures used for SFD [83]. Drying methods are of great importance to the residual moisture content of the products [1]. A high enough water content not only affects the physical characteristics of the produced particles, as water decreases the Tg, resulting in a less stable solid structure [54], but it may also affect the stability of the active ingredients such as an antigen [31]. Furthermore, in food applications, moisture content is a crucial parameter, determining factors such as shelf life as it affects mold growth and agglomeration [82]. The reduction of the water content below a certain threshold is critical for ensuring product acceptability [89].

Other drying techniques, such as FD, often require an additional milling step for the production of a uniform powder [103]. By employing SD, this step can be avoided, as well as the investment in specialized cryogenic equipment [45]. Nevertheless, certain limitations must be considered, including particle shrinkage during moisture evaporation, low thermal efficiency that results in high energy consumption, low yield at the laboratory scale due to product deposition on the drying chamber wall, and high maintenance costs because of nozzle clogging. Furthermore, SD is not suitable for thermoplastic and hygroscopic materials, as well as for expensive pharmaceutical products, because it is impossible to retrieve 100% of the material from these types [26].

### 6.2. Comparison with FD

FD overcomes the challenge of preserving heat-sensitive components. It consists of two drying stages: during the primary stage, the moisture is removed via sublimation, and during secondary drying, the remaining water is removed by desorption [2]. Primary drying is usually the most time-consuming step, taking from a few days to weeks, whereas secondary drying occurs over several hours [26]. Conventional vacuum FD requires a long time due to poor heat and mass transfer rates, depending on the sample thickness [87]. It is a batch process that has low energy efficiency and consequently has high operational and capital costs [1,13,14,15]. The resultant solid formulations have a cake-like, porous structure and typically require further processing, such as milling, to form powders for needle-free ballistic injections or inhalation treatments [15,28].

FD can be utilized for microencapsulation and the manufacturing of high-quality products, with remarkable rehydration behavior due to their porous structure and physical and chemical stability [1,83,104]. However, conventional FD process parameters, such as product temperature, shape, size, and sterile process conditions, are challenging to control [29,87].

Although FD operates at lower temperatures than SD, helping to prevent heat damage to particles, the lyophilization process can still introduce stresses that contribute to material degradation [15,104]. Low temperatures often cause protein unfolding (cold denaturation), while ice formation can lead to concentration of the solutes (freeze concentration or cryo-concentration), which could cause the separation of the amorphous phases and/or crystallization of components, something that may compromise the product’s stability [15,26,104]. To overcome these limitations, various excipients or cryoprotectants can be utilized, often selected through trial and error. Cryoprotectants either trap the particles in a glassy matrix or function as water replacements, forming hydrogen bonds between them and the nanoparticles [104]. It is important to recognize that different excipients are effective for different techniques. For instance, while mannitol is widely used for SFD, it can cause instability in SD and FD products because it crystallizes during processing and storage [15].

The main drawback of FD is the long processing time. SFD addresses this challenge by reducing the particle size, which decreases the process duration and energy consumption and contributes to faster freezing and drying rates [1,104]. Due to the rapid freezing process, drug solutions can be amorphously embedded in the excipients, reducing the potential of phase separation and resulting in a uniform molecular distribution [27]. Faster freezing rates promote the formation of dendrite-type crystals with thin interstitial spaces, which improves nanoparticle stability by reducing mechanical stresses and preventing aggregation [104]. Additionally, using less humid and lower temperature drying mediums than those used in SD produces uniform microstructures and minimizes aroma compound losses [2,27,105].

SFD is a promising technique for producing porous powders with improved flowability and for encapsulating heat-sensitive materials [82]. The generated particles showcase excellent rehydration behavior, facilitated by their high porosity, and they maintain their spherical shape that was achieved during atomization due to rapid evaporation rates, as opposed to the SD-produced particles, which may undergo shrinkage during drying [102,105]. FD allows for the production of free-flowing powders with regulated particle size distribution, which is largely determined by the atomization process, while the low residual water content of the final product enhances their stability, allowing storage at room temperatures [26,27,105].

In pulmonary drug delivery, particle size, morphology, and density are of major significance for optimizing lung deposition and therapeutic efficacy [106]. As mentioned previously, SD produces smaller, denser particles that are appropriate for fine-tuning aerodynamic properties [107]. Polymers like poly(lactic-co-glycolic acid) (PLGA) and carbohydrates, such as mannitol and lactose, are commonly used in SD formulations to enhance stability and control drug release [108]. In contrast, SFD produces larger particles with high porosity, reduced density, and a spherical structure, enhancing dispersibility and lung deposition due to favorable aerodynamic characteristics. These can also reduce particle aggregation and enhance lung distribution, making it an appealing technique to produce powders that target specific pulmonary regions [2].

Employing SFD or SD to alter particle size and drug release rates of drug formulations can have a direct impact on their efficacy. For example, Ananya et al. (2025) demonstrated how effective polymer-based solutions with anti-inflammatory properties are in pulmonary drug delivery. The study proposes the combination of heparin and azithromycin to reduce inflammation and inhibit pathogens such as SARS-CoV-2 and bacteria associated with lung infections [109]. De Pablo et al. (2023) explored the use of SD to develop dry powder formulations that can target lung macrophages for treating fungal and peracetic infections. The authors highlighted how particle engineering can help optimize drug delivery to the deep lung while enhancing macrophage uptake [110]. The use of innovative excipients may further improve lung deposition by controlling particle size and density through the atomization process. By carefully selecting polymers or carbohydrates in SD and SFD, controlling the particle size and density, the enhancement of drug targeting within the lung, and the improvement of therapeutic efficacy can be achieved.

The combination of SD and FD techniques could optimize product quality and stability while also broadening the range of feed acceptability and expanding the end uses for the dried products [111]. Potential applications include biological entities, high-value foods and drugs, substances that are susceptible to environmental factors, and the production of dry powder inhalations that are administered through the nose, as their small aerodynamic diameters make them suitable for pulmonary delivery [10,102,104,111]. Moreover, during FD, the liquid is often placed into trays or vials, which complicates the even distribution of heat. In contrast, SFD achieves homogeneous exposure of the product to the process conditions, promoting efficient moisture sublimation rates and providing control over vial dose, increasing production flexibility [27,111]. SFD can also enhance the dissolution rate of poorly water-soluble drugs and achieve low residues of organic solvents [27,29].

Although SFD minimizes processing time and limits the cold denaturation of the materials in comparison with conventional FD, protein unfolding and aggregation may still be evident [28,49]. Additionally, the low density of the particles can cause issues in industrial processing, and high surfactant concentrations can reduce surface tension between the droplet and drying medium, resulting in smaller, less spherical particles [104].

Selecting the optimum technique and process conditions depends on the desired product properties. Elik et al. (2021), while encapsulating flaxseed oil, rich in carotenoids, observed superior flow properties on the powders prepared with SFD but lower encapsulation compared to the SD particles [82]. Deotale et al. (2020) examined the effects of SD, FD, and SFD during the drying of roasted arabica coffee beans. Higher yield was achieved with the SFD process (38%), followed by FD (23%) and SD (11%), showcasing the importance of the freeze-drying step. Furthermore, while agglomeration was evident in SD particles, they had lower residual moisture. SD and SFD resulted in smooth, spherical particles, with the latter having excellent flowability and dissolution behavior [92].

H. Yu et al. (2016) utilized SD and SFD for the formulation of a carrier-free DPI of antibiotic nanoplex intended for bronchiectasis treatment. Their results showed that L-leucine content affected the shape of the SD-produced powders but had no effect on SFD powders. Additionally, it did not affect the aerosolization efficiency of the particles produced with SD, but it did have an impact on the SFD powder size distribution [68]. Karthik & Anandharamakrishnan (2013), while microencapsulating DHA using all three methods, achieved greater encapsulation efficiency but increased oxidation during SD. Although SFD produced spherical powders with good rehydration behavior and stability, it also achieved the lowest yield [83].

During the drying of pharmaceutical proteins with FD and SFD, Leuenberger et al. (2006) concluded that the latter showed shorter drying times and better control of drying parameters but also resulted in lower yield during the primary drying and strong electrostatic effects during the secondary drying [29]. SFD also caused aggregation of darbepoetin alfa during drying experiments of Nguyen et al. (2004), which could be mitigated by annealing before drying. SD did not have an impact on the particle’s integrity and resulted in a uniform size distribution [45]. Finally, Y. Zhang et al. (2014) used SFD to study the synthesis of nanozirconia granules and demonstrated the significance of the atomizing step. SD ceramic powders exhibited high strength and were difficult to crush at standard industrial pressing pressures, which resulted in uncrushed granules that reduced the strength of the sintered components. Some of the SFD-prepared granules had low flowability and yield, while some had low density and/or poor strength due to high density or uneven structure. The researchers addressed these issues by employing two fluid atomizers and ultrasonic SFD [40].

The differences in the final structure of SFD-, SD-, and FD-prepared particles can be better understood in Figure 6, which consists of SEM pictures of micellar casein powder prepared with SFD (Figure 6a,b), SD (Figure 6c,d) and FD (Figure 6e,f). The SFD particles had a spherical shape and high porosity, while the SD produced smooth spherical microcapsules, in which particle shrinkage is evident. Figure 6e,f show the cake-like structure of the FD powder with a porous surface [90].

A thorough comparison of SFD with SD and FD, taking into account the significant factors of operation requirements, scalability, cost of equipment and operation, and the characteristics of the final products, stability, particle size, etc., is demonstrated in Table 5.

## 7. Future Needs

Although significant advances have been made toward a more efficient and scalable SFD, this technology remains not well established in the industry. The majority of the SFD technologies are designed to operate in batch mode. However, with small adjustments, they can be converted to continuous, reducing the costs associated with long batch cycles. For example, Rey’s concept proposes a continuous SFD process in which the atomized particles are frozen in a counter-current cold air stream. Sublimation occurs on heated vibrating belts, and the dried product is continuously poured into the vials in the vial section [5].

Meridion Technologies has developed a different SFD setup that consists of a spray freezing chamber (SprayCon) and a rotary drum (LyoMotion), which can produce particles with diameters ranging from 250–800 μm [5]. Although LyoMotion is the first industrial-scale freeze–dryer that has been tested as a batch technology, it has the potential to work continuously with the appropriate system for the discharge of the dried particles. Other continuous designs include (i) the fine-SFD from ULVAC Technologies, which is still under development regarding its transition from a batch process, (ii) the Stirred FD (or active FD) based on Hosokawa Microns’s concept, and (iii) LYnfinity which is an improved version of Rey’s concept, introduced by the IMA Group. Transforming SFD into a continuous process would enhance flexibility, process control, and scalability [1,3,5].

Moreover, the FD step of the SFD process leads to significant capital and operational costs. This is primarily due to the low processing temperatures, the requirement for vacuum conditions, or large amounts of dry cold gas in the case of atmospheric SFD, all of which jeopardize the economic viability of the SFD operation. Processing costs for SFD are estimated to be 30 to 50 times higher than conventional SD [51]. Atmospheric FD has been proposed, but materials with low eutectic or Tg demand extremely low drying temperatures, driving up the cost per kilogram of dried powder. Furthermore, most pharmaceuticals and food products have indeterminate eutectic and Tg, rendering SFD prototypes unsuitable for use [1]. As a result, more in-depth research into the effects and optimization of the SFD process in the pharmaceutical and food industry is needed. A thorough understanding of factors that affect freezing and drying rates, droplet size, and geometry is essential to optimizing the process parameters and enhancing the product’s quality [105].

SFD is an effective method for drying a wide range of foods and biological compounds, but due to its high costs and the limited economic viability of the food industry, its application mainly focuses on the production of high-value pharmaceutical products and foods with bioactive or volatile components. Recent developments in SFD technology have improved process control, reduced drying times, and enhanced the viability of volatiles, allowing for the production of higher-quality products. Further comprehensive research on physicochemical, textural, and sensory characteristics is necessary to ensure the possibilities of the SFD technology. Furthermore, this method enables the production of pharmaceutical powders suitable for nasal, pulmonary, and needle-free epidermal applications, opening new research avenues for understanding the mechanism of SFD particles in drug delivery in the human body. Moreover, future research could focus on the encapsulation of proteins, essential oils, probiotics, and aroma volatiles to meet both commercial and research needs [111].

The industrialization of the SFD process faces many challenges that need to be overcome. One major issue is the design of SD and FD chambers so that they can produce powders of different sizes. Furthermore, the use of cryogens, such as liquid nitrogen, requires extra safety measures in terms of operational safety and economic viability. Significant amounts of cryogens are required in the SFD process, highlighting the need for further research on minimizing waste. On the other hand, industrial and regulatory aspects such as qualification, process validation, good manufacturing practice, scale-up, and scale-down potential, as well as purchase, operation, and energy consumption costs, have not been thoroughly addressed. Additionally, the sterility of the process and the precise dosage requirements essential for pharmaceutical manufacturing must be ensured [2].

Beyond the economic considerations of the SFD process, there are additional technical issues that have to be resolved. The atomization step increases the liquid-gas contact area, leading to protein unfolding and adsorption. Furthermore, rapid freezing can cause protein denaturation, crystal nucleation, and growth, which in turn slows the freezing process, resulting in bigger particle sizes and lower interfacial areas. The elutriation of fine powders further complicates the process [105]. Additional research on these topics is required to fully exploit this technology’s potential on an industrial scale.

## 8. Conclusions

SFD has gained remarkable popularity as a technology that bridges the benefits of conventional SD and FD methods, producing highly porous, stable powders suitable for pharmaceuticals, foods, and other specialized industrial applications. The combination of rapid freezing followed by controlled sublimation offers a solution for preserving thermosensitive compounds and achieving exceptional quality in the final products.

However, SFD faces significant barriers of an economic and technical nature, which need to be overcome to ensure that the technique becomes widely adopted within the industry. The high initial investment costs, along with the complicated operational requirements, limit its scalability. In order to broaden the potential of SFD and make it industrially applicable, advancements in the process design, optimization, and modeling, along with the incorporation of the three steps of SFD into a single, sterile, continuous process, is deemed to be an important hurdle to fully utilizing the benefits of this technology.

Future research should focus on optimizing the process parameters, improving product quality, and expanding the range of applications. As the technology evolves, overcoming these challenges will establish SFD as a leading method for producing high-performance powders in diverse sectors, paving the way for breakthroughs that have the potential to completely modify product composition and delivery system.

## Figures and Tables

**Figure 1 pharmaceutics-16-01533-f001:**
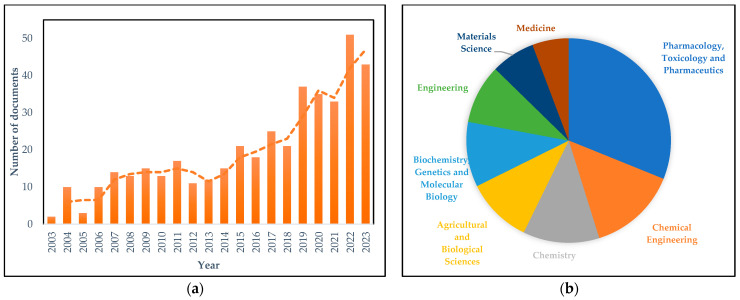
(**a**) Number of publications working on the technology of SFD during the last 20 years and (**b**) subject areas of SFD, according to Scopus data.

**Figure 2 pharmaceutics-16-01533-f002:**
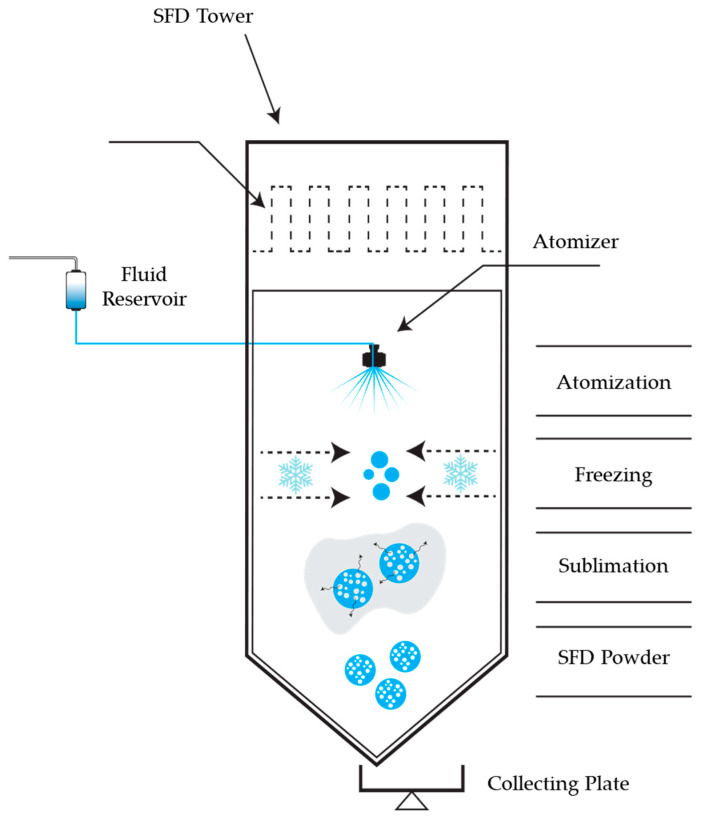
Illustrations of the SFD process and the sub-processes of atomization, freezing, and drying (sublimation).

**Figure 3 pharmaceutics-16-01533-f003:**
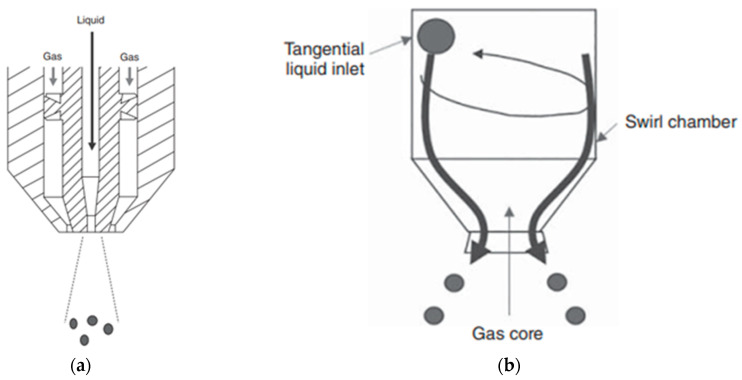
Configurations of (**a**) two-fluid atomization nozzle, (**b**) hydraulic atomizer [26] (reproduced with permission from Ohtake et al., copyright John Wiley and Sons).

**Figure 4 pharmaceutics-16-01533-f004:**
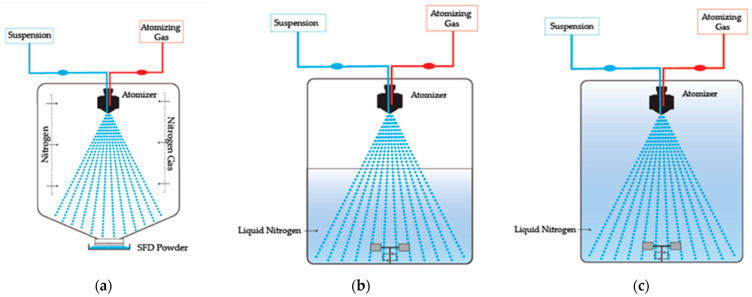
Freezing configurations: (**a**) spray freezing into vapor, (**b**) spray freezing into vapor over liquid, and (**c**) spray freezing into liquid.

**Figure 5 pharmaceutics-16-01533-f005:**
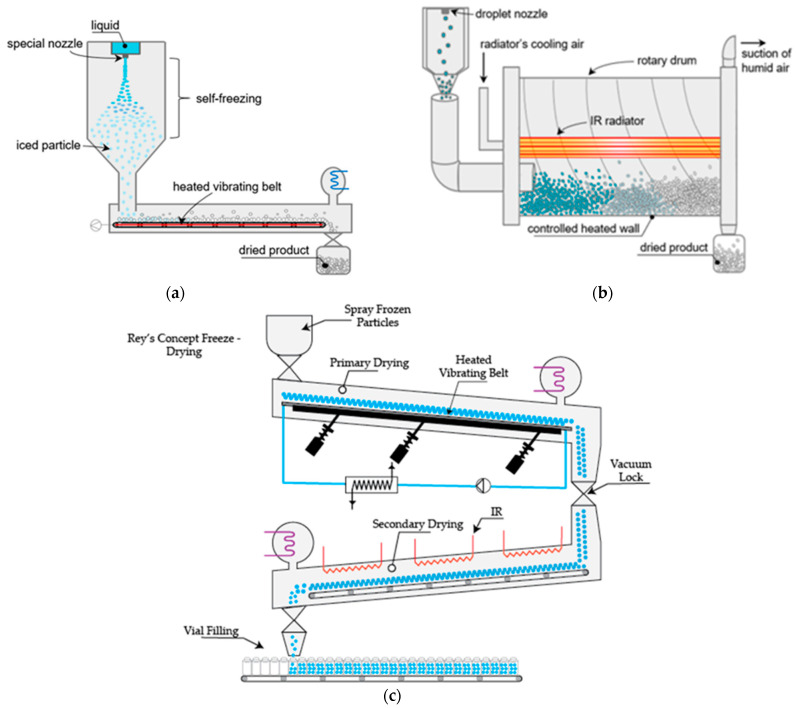
Configurations of (**a**) fine-SFD system [5], (**b**) dynamic FD [5], and (**c**) Rey’s concept.

**Figure 6 pharmaceutics-16-01533-f006:**
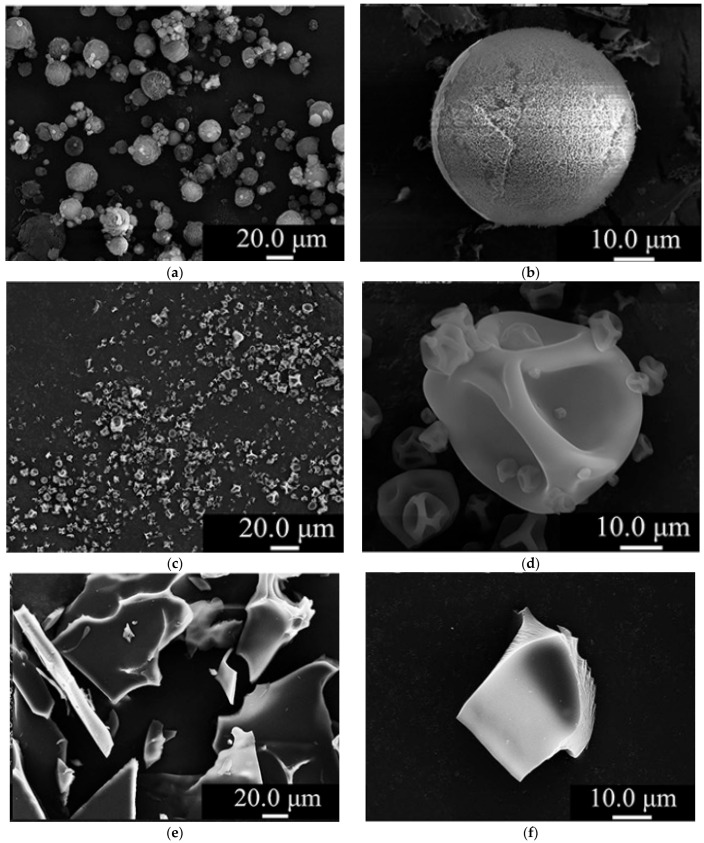
Powder morphology of (i) spray-freeze-dried at (**a**) at 500× and (**b**) 1500× magnitude, (ii) spray-dried at (**c**) at 500× and (**d**) 1500× magnitude and (iii) freeze-dried at (**e**) at 500× and (**f**) 1500× magnitude micellar casein [90] (reproduced with permission from Ren et al. copyright Elsevier).

**Table 1 pharmaceutics-16-01533-t001:** Significant adjustable parameters during SFD and their effect on the characteristics of the final products.

Parameter	Effect	Reference
Solution concentration ↑	Particle size ↑	[30]
	Particle density ↑	[31]
	Residual moisture ↓	[30]
	Porosity ↓	[32]
	Sphericity ↑	[8]
Viscosity ↑	Particle size ↑	[11]
	Yield ↓	[6]
Nozzle diameter ↑	Particle size ↑	[30]
Hydraulic nozzle liquid feed flow rate ↑	Particle size ↓	[33]
Two-fluid nozzle atomizing gas flow rate ↑	Particle size ↓	[34]
	Yield ↓	[35]
Two-fluid nozzle liquid feed flow rate ↑	Particle size ↑	[36]
	Yield ↑	[35]
Use of four-fluid nozzle	Particle size ↓	[37]
	Porosity ↑	[37]
Use of ultrasonic nozzle	Particle size distribution ↓	[4]
Ultrasonic nozzle frequency ↑	Particle size ↓	[4]
Primary drying temperature ↑	Particle size ↓	[4]
	Drying time ↓	[38]

↑ indicates an increase; ↓ indicates a decrease in the values of the parameters and their effects.

**Table 2 pharmaceutics-16-01533-t002:** Applications of SFD for the development of pharmaceuticals through atomization.

	Lung Diseases				
Purpose	Core Material	Type Of Final Product	Particle Characteristics	Target Application	Reference
Development of site targeting MTK microparticles for respiratory drug delivery	Montelukast sodium (MTK)	Powder for pulmonary delivery	D50: 4.95–16.56 μm FPF: 60%	Asthma treatment	[60]
Preparation and evaluation of ivacaftor–colistin co–loaded nanosuspension	Colistin and ivacaftor	Powder for pulmonary delivery	d50: 170.6–251.9 μm FPF: 63.6%	Treatment of lung infections caused by Pseudomonas aeruginosa	[7]
Production of sildenafil citrate porous microparticles for pulmonary delivery	Sildenafil citrate	Powder for pulmonary delivery	d50: 8.27 μm FPF: 25.53% Yield: 49.30% Entrapment efficiency: 94.48% Residual moisture content: 6.38%*w*/*w*	Treatment of pulmonary arterial hypertension	[6]
Production of hyaluronic acid coated liposome–protamine–DNA complex	siRNA	Powder for pulmonary delivery	d50: 29.4 μm FPF: 39.2%	Treatment of lung diseases	[52]
Evaluation of the effects of process parameters and formulation composition on voriconazole powder properties	Voriconazole	Powder for pulmonary delivery	d50: 2.2–11.6 μm FPF: 44% Drug content: 33.0–74.3%	Mycosis treatment	[76]
Production of voriconazole powder for pulmonary delivery	Voriconazole	Powder for pulmonary delivery	MMAD: 2.78–5.89 μm FPF: <40% Drug content: 19.2–43.2%	Treatment of pulmonary aspergillosis	[8]
Preparation of amino acid–based stable adalimumab formulation	Adalimumab	Powder for pulmonary delivery	d50: 8.10–14.99 μm FPF: 25.2–67.5%	Asthma treatment	[67]
Preparation of DPI formulation of siRNA	siRNA	Powder for pulmonary delivery	d50: 8.65–14.89 μm FPF: 28% Yield: 83.70–93.76	Treatment of lung diseases	[53]
Production of siRNA powder for pulmonary delivery	siRNA	Powder for pulmonary delivery	d50: 5.61–7.22 μm FPF: 6.55–42.2%	Treatment of lung metastasized tumors	[64]
Production of siRNA powder for pulmonary delivery	siRNA	Powder for pulmonary delivery	d50: 6.02–13.04 μm FPF: 30.4% Yield: 92.63–99.18%	Treatment of lung diseases	[35]
Improvement of aerosolization efficiency of clarithromycin liposomal dry powder	Clarithromycin (CLA)	Powder for pulmonary delivery	d50: 0.3700–0.6936 μm FPF: 50% Drug content: 85.02–92.14%	Treatment of lung diseases	[55]
Production of ciprofloxacin nanoplex for pulmonary delivery	Ciprofloxacin	Powder for pulmonary delivery	MMAD: 3 μm FPF: 29%	Bronchiectasis treatment	[68]
Production of terbutaline sulfate powder for pulmonary delivery	Terbutaline sulfate	Powder for pulmonary delivery	MMAD: 4 μm FPF: 22.9%	Asthma treatment	[42]
Production of salmeterol xinafoate powder with enhanced physical characteristics and dissolution rate	Salmeterol xinafoate	Dry powder	d50: 10.9 μm	Treatment of asthma symptoms	[69]
	**Vaccines**				
**Purpose**	**Core material**	**Type of final product**	**Particle characteristics**	**Target Application**	**Reference**
Micropellet formulation for yellow fever vaccine	Yellow fever virus (vYF)	Stabilized vYF micropellets	d50: 528–584 μm Residual moisture: 0.8–2.5%	Yellow fever vaccine	[58]
Production of influenza vaccine for pulmonary administration	Whole inactivated virus	Powder for pulmonary delivery	d50: 5.497–8.660 μm	Influenza vaccination	[70]
Stabilization of hepatitis B vaccine	Hepatitis B surface antigen	Dry powder	d50: 0.0201 μm Residual moisture: 2.59%	Vaccination against HBV	[54]
Stabilization of influenza subunit vaccine powder	Influenza monovalent vaccine (mainly composed of haemagglutinin glycoprotein)	Powder for pulmonary delivery	d50: 2.61–10.87 μm FPF: 23%	Pulmonary vaccination against the flu	[10]
Production of influenza vaccine for pulmonary delivery	Monovalent influenza subunit vaccine	Powder for pulmonary delivery	d50: 10.36–24.57 μm	Influenza vaccination	[9]
Production of influenza vaccine for epidermal delivery	Monovalent influenza vaccine	Powder for epidermal delivery	d50 35–56 μm Yield: 82–89% Residual moisture: 1.5–3.0%	Influenza vaccination	[31]
	**Antibiotics**				
**Purpose**	**Core material**	**Type of final product**	**Particle characteristics**	**Target Application**	**Reference**
Production of porous celecoxib lyospheres with increased bioavailability	Celecoxib	Capsules for oral delivery	d50: 213.44–793.10 μm	Increased drug absorption of poorly soluble celecoxib	[11]
Enhancement of Azithromycin dissolution and bioavailability	Azithromycin	Dry powder	d50: 54.44 μm Drug content: 98.48%	Improved Azithromycin antibiotic effectiveness	[72]
Production of ciprofloxacin powder for pulmonary delivery	Ciprofloxacin	Powder for pulmonary delivery	MMAD: 2.8 μm Encapsulation efficiency: 50–93.5%	Pulmonary delivery of antibiotics	[57]
	**Other**				
**Purpose**	**Core material**	**Type of final product**	**Particle characteristics**	**Target Application**	**Reference**
Production of physically stable pharmaceutical protein solids	Bovine serum albumin (BSA)	Dry powder	d50: 35.1–59.5 μm Residual moisture: 4.68–6.73%	Pharmaceutical applications	[50]
Investigation of the influence of different preparation methods on the structure of low—soluble SFD nanoparticles	Mometasone furoate (MF), moterol fumarate (FF)	Powder for pulmonary delivery	d50: 1.72–3.43 μm FPF: 35.22–49.10%	Pharmaceutical applications	[59]
Production of ointment containing micronized metronidazole	Metronidazole	Ointment	d50: 2.7 μm	Treatment of symptoms of rosacea	[56]
Examination of DMSO as a novel solvent for poorly soluble SFD drug powders	Rivaroxaban	Powder for pulmonary delivery	d50: 250–350 μm DMSO residual content: 2.4–5.4%	Pharmaceutical applications	[71]
Nanoencapsulation of antioxidant peptides	*Lupinus* *mutabilis* proteins	Nanoparticles for colonic delivery	d50: 332–465 μm Encapsulation efficiency: 63.80–71.75%	Treatment of inflammatory bowel disease	[61]
Investigation of intranasal drug delivery and controlled release of resveratrol	Resveratrol	Powder for intranasal delivery	d50: 171.52–178.92 μm Residual moisture: 1.09–5.66%	Treatment of CNS disorders	[62]
Preparation and evaluation of inhalable microparticles of Rizatriptan benzoate	Rizatriptan benzoate	Powder for pulmonary delivery	d50: 3.17–9.74 μm FPF: 17.54–61.10%	Treatment of acute migraine headache	[73]
Stabilization of immunoglobulin G particles	Immunoglobulin G (IgG)	Dry powder	-	Therapeutic antibodies	[77]
Production of microparticles with increased enzymatic activity and stability	Trypsin	Dry powder	d50: 130–161 μm	Manufacturing of dry powders that retain biological activity	[28]
Use of hydroxypropylcellulose as a matrix	Diterpenoid lactone	Dry powder	-	Improved dissolution and bioavailability of poorly soluble drugs	[78]
Production of protein powder for epidermal delivery	Bovine serum albumin or bovine carbonic anhydrase	Powder for epidermal delivery	-	Medical applications	[75]
Production of porous inhalable parathyroid hormone	Parathyroid hormone (PTH) (1—34)	Powder for pulmonary delivery	d50: 14.07–16.02 μm FPF: 14.9–91.1%	Treatment of osteoporosis	[74]
Encapsulation of nanoparticles for pulmonary delivery	Cholesterol nanoparticles	Powder for pulmonary delivery	d50: 21 μm FPF: 60% Drug loading: 50%	Pulmonary delivery of therapeutic nanoparticles	[79]
Dry powder inhaler formulation of drug–loaded lipid–polymer hybrid nanoparticles	Levofloxacin	Powder for pulmonary delivery	d50: 15–17 μm FPF: 2–26% Yield: 20–47%	Pulmonary delivery of therapeutic nanoparticles	[66]
Increased oleanolic acid dissolution rate and bioavailability	Oleanolic acid	Dry powder	Residual content of butan-1-ol: 0.48–10.57 ppm	Treatment of hepatitis	[63]
Production of insulin powder for ballistic delivery	Insulin	Powder for ballistic delivery	d50: 2.78–48.45 μm Drug content: 9.9–44%	Improved delivery method of insulin	[4]
Stabilization of THC	Δ9-tetrahydrocannabinol (THC)	Powder for pulmonary delivery	d50 3.45 μm FPF: 35–50% Drug content: 4–30%	Pulmonary administration of THC	[18]
Encapsulation of darbepoetin alfa	Darbepoetin alfa	Dry powder	d50: 29 μm Yield: 73–82% Encapsulation efficiency: 95%	Protein drug for prolonged red blood cell mass regulation	[80]
Production of powder for epidermal delivery	Trypsinogen	Powder for epidermal delivery	d50: 20–80 μm	Manufacture of drug powders for epidermal delivery	[25]
Production of protein aerosol powder	Recombinant-derived humanized anti–IgE monoclonal antibody (Mab) and recombinant human deoxyribonuclease (rhDNase) proteins	Powder for pulmonary delivery	d50: 8–10 μm FPF: 50–70%	Biopharmaceutical powder in drug delivery systems	[81]

**Table 3 pharmaceutics-16-01533-t003:** Applications of SFD for the development of foods and supplements.

Purpose	Core Material	Type of Final Product	Final Product Characteristics	Target Application	Reference
Production of micellar casein powders with controlled droplet size and good solubility	Micellar casein	MC powder	d50: 20.6–86.4 μm Moisture content: 2.83–3.44%	Nutritional supplements	[90]
Enhancement of solubility and redispersibility of SFD dairy powders	Cow and goat milk	Dairy powder	Moisture content: 4.6–10.9%	Dairy products	[91]
Production of encapsulated carotenoid-enriched flaxseed oil	Carotenoid-enriched flaxseed oil	powder	d50: 38.97 μm Moisture content: 1.13% Encapsulation Efficiency: 43.5–70.9%	Increased stabilization of antioxidants	[82]
Encapsulation of *Saccharomyces cerevisiae*	*Saccharomyces cerevisiae*	Enzyme powder	-	Increased stability of *Saccharomyces cerevisiae* and superior quality ice wine production	[89]
Production of superior instant coffee foam	Concentrated coffee extract	powder	d50: 0.329 μm Moisture content: 6.72% Yield: 37.92%	Improved instant coffee foam	[92]
Encapsulation of microbial transglutaminase	microbial transglutaminase	powder		Increased stability for food applications	[14]
Enhancement of bioavailability of vitamin E with encapsulation	Vitamin E	Vitamin E microcapsules	d50: 145.3 μm Encapsulation efficiency: 89.3% Moisture content: 5.41%	Functional food	[88]
Encapsulation of Lactobacillus plantarum	*Lactobacillus plantarum*	powder	d50: 53.99–105.07 μm Moisture content: 5.57–6.32% Encapsulation efficiency: 87.92–94.86%	Microencapsulation of probiotics	[87]
Production of encapsulated vanillin with increased stability	Vanillin	powder	d50: 24.76–165.40 μm Moisture content: 4.15–6.63% Encapsulation efficiency: 72%	Food flavorings	[13]
Preparation of probiotic powder	*Lactobacillus casei*	Fine probiotic powder	D50: 24.8	Functional food for intestinal microbial balance	[84]
Encapsulation of Docosahexaenoic Acid (DHA) for increased stability	DHA	powder	Moisture content: 3.667% Encapsulation efficiency: 70.77%	Functional foods	[83]
Microencapsulation of *Lactobacillus plantarum*	*L. plantarum*	Microencapsulated *L. plantarum* powder	Moisture content: 3.48–3.59%	Functional food for intestinal microbial balance	[86]
Microencapsulation of *Lactobacillus paracasei*	*L. paracasei*	Dry microcapsules of *L. paracasei*	Moisture content (dry basis): 0.09–0.15	Functional food for intestinal microbial balance	[85]

**Table 4 pharmaceutics-16-01533-t004:** Applications of SFD for the development of catalysts, aerogels, insulating, and other materials.

Purpose	Core Material	Type of Final Product	Final Product Characteristics	Target Application	Reference
Production of composite catalyst	TiO_2_ or SiO_2_	Composite material powder	Ethylene productivity: 0.36–0.22 g_ethylene_/g_aquivision_/min Selectivity: 93–89%	Catalysts	[93]
Production of PtCu_3_/rGO nanoparticles via a traceless protectant	PtCu_3_/rGO	PtCu_3_ nanoparticles	D_avg_ = 2.7 nm 6.5–5.8 times higher MA and SA than commercial Pt/C	Oxygen reduction reaction catalysts	[95]
Investigation of the influential parameters on the granulation process of spinel powders	Magnesium aluminate spinel (MgAl_2_O_4_)	Spinal powder	d = 10–50 μm	Oxide spinels and ceramic materials	[96]
Preparation of nanostructured porous mixed oxides	TiO_2_, Pt	Porous micro—granules	HMF conversion: 98%, BHMF selectivity: 100%	Application in catalytic processes	[94]
Preparation of monolithic silica aerogels	Silica	Silica aerogels	Thermal conductivity: 0.0215 W/mK	Thermal insulation materials	[99]
Production of transparent MgAl_2_O_4_ spinel	MgAl_2_O_4_	Hot isostatically pressed surface	D_50_ = 150 nm	Manufacture of highly transparent ceramics	[97]
Development of superinsulating bioaerogels	Nanofibrillated cellulose (NFC)	Nanofibrillated cellulose aerogels	Thermal conductivity: 0.018 W/mK S_BET_ = 80–100 m^2^/g	Insulating applications	[98]
Examination of SFD nanosized silicon carbide that contains granules	Silicon carbide	Spherical granules comprising silicon carbide nanoparticles	D_avg_ = 40 nm	Ceramic materials	[100]
Production of FOX-7 three-dimensional grid out of nanostructures	1,1-Diamino-2,2-dinitroethylene (C2H4N4O4, FOX-7)	quasi-three-dimensional grid of one-dimensional nanostructure	D = 50–200 μm V_50_ = 13.19 kV E_50_ = 2.65 J	Production of high energy density materials	[101]

**Table 5 pharmaceutics-16-01533-t005:** Comparison of the drying processes of SFD, SD, and FD regarding the operation characteristics: scalability, equipment and operation cost, and the product characteristics: particle size, stability, etc.

Factors	Spray Freeze Drying (SFD)	Spray Drying (SD)	Freeze Drying (FD)
Operation	Needs specialized equipment [2] Complex process [111]	Needs specialized equipment [2]	Needs specialized equipment [2] Time—consuming [111]
Scalability	Mostly batch, a few continued concepts have been proposed [111]	Continuous, easy to scale up [14,62,83]	Mostly batch [111]
Cost of equipment	High capital costs [105]	Low cost [83]	High capital costs [83]
Cost of operation	Lower operation costs than FD, as it acquires less time and energy consumption [27]	Low operational costs [14]	High running costs because of the energy-intensive operation needed by vacuum or batch operation [2,105] Expensive [83,111]
Particle size	Close to original droplet size, which depends on the atomizer [112] No shrinkage during drying [62]	Various sizes, from microns to submicrons Shrinkage of droplets during evaporation [2]	Micron size [2]
Stability	Good stability as it maintains product quality [2]	-	Improved stability and shelf-life duration [2]
Product properties	Small, porous particles, smaller density than SD [62] large surface area Good flowability Good dissolution behavior [105]	Bigger, spherical, non-porous particles Lack of homogeneity [2]	Irregular forms, porous, low density [2]

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
