# Peer review of "A Comprehensive Review of the Latest Trends in Spray Freeze Drying and Comparative Insights with Conventional Technologies"

_pharmaceutics, 2024, doi:10.3390/pharmaceutics16121533_

Round 1
Reviewer 1 Report
Comments and Suggestions for Authors
Dear Authors,
Your manuscript is very relevant and valuable for researchers and professionals in the fields of pharmacy and food technology. The focus on the innovative drying technology (SFD) is especially significant, as it is used to produce medicines, food products, and active ingredients with improved quality and stability. I have a few comments that might help refine your work: Could you specify how many scientific articles were chosen and analyzed during the preparation of this study? Additionally, was every article you found included, or did you use specific criteria to select them? If so, it would be useful to explain these criteria.Also suggest giving a more detailed description of the results shown in Figure 1 in the introduction to help readers better understand the data. In the introduction, it might be helpful to delve deeper into the importance of your chosen topic. Despite the economic and technical hurdles associated with SFD, why is it essential to explore its potential in the pharmaceutical and food industries? Clarifying the objective of your study would also enhance the manuscript by providing readers with a clearer sense of its purpose and contributions. Lastly, there seems to be limited information on the benefits of SFD for the quality of pharmaceutical products, especially how it helps improve the bioavailability of active components. Expanding on this point would add valuable context in Introduction.
Author Response
Reply to Reviewer 1
We would like to thank the Reviewer for the useful comments and suggestions made on our manuscript, which helped us to improve it substantially. Please find below the changes made on our manuscript following the Reviewer’s recommendations.
Your manuscript is very relevant and valuable for researchers and professionals in the fields of pharmacy and food technology. The focus on the innovative drying technology (SFD) is especially significant, as it is used to produce medicines, food products, and active ingredients with improved quality and stability. I have a few comments that might help refine your work:
- Could you specify how many scientific articles were chosen and analyzed during the preparation of this study?
During the preparation of the review, more than 150 papers were chosen and studied. However, some of them, were not included in our manuscript either because they referred to very old and “outmoded” results or because they did not include an innovation that is worth to mention. In our manuscript, we have referenced over 100 papers to provide a comprehensive overview of the topic. We carefully selected these references to cover a wide range of studies and to ensure that our discussion is well-grounded in the field's recent advancements and pharmaceutical applications.
The procedure for the preparation of the review was firstly to exhaust the literature finding all the available papers with the topic of SFD and secondly storing them properly in a shared folder. After that, each article was analysed according to its research topic and focus. For example, the research papers where their potential use was mentioned, were included in the Section of Application. Those which provided useful info with the comparison of the techniques were presented in the respective Section etc.
- Additionally, was every article you found included, or did you use specific criteria to select them? If so, it would be useful to explain these criteria.
After extensive research, most of the published papers on SFD, from the available bibliography were included. For the tables on applications, the criteria for selecting the presented works were whether a specific application or a possible potential application was analysed in the research. For the rest of the sections, we chose to present research that inquired into the process by which SFD affects the final product characteristics.
- Also suggest giving a more detailed description of the results shown in Figure 1 in the introduction to help readers better understand the data.
A more detailed description of the data in Figure 1 is provided in the text
- In the introduction, it might be helpful to delve deeper into the importance of your chosen topic.
Several respective paragraphs are added in the Section of Introduction. Please inform us if anything else is needed.
- Despite the economic and technical hurdles associated with SFD, why is it essential to explore its potential in the pharmaceutical and food industries? Clarifying the objective of your study would also enhance the manuscript by providing readers with a clearer sense of its purpose and contributions.
We would like to thank the Reviewer for this insightful note. We added the following paragraph highlighting the objective of the review in the pharmaceutical and food industries in the Section of Introduction l.39: “Selecting the appropriate drying process can prove to be a challenge when dealing with thermosensitive compounds. SFD is not only appropriate for such substances but also as a gentler technique it minimizes processing losses, maintains the product integrity and adds distinct product qualities (enhanced hydration, porous structure, rapid reconstitution, etc.). These characteristics offer this technique a competitive edge over conventional drying methods, making it suitable for the processing of various drugs, water soluble or not, production of powders with specific characteristics, and the encapsulation of bioactive compounds in drugs and functional foods.”
- Lastly, there seems to be limited information on the benefits of SFD for the quality of pharmaceutical products, especially how it helps improve the bioavailability of active components. Expanding on this point would add valuable context in Introduction.
We would like to thank the Reviewer for this helpful comment. We provided a paragraph, where we clearly state how SFD improves the bioavailability of active compounds. The first paragraph is found in the Introduction Section l.47:” SFD has gained attention for its ability to enhance drug bio-availability, stability, and delivery efficiency, particularly for poorly soluble compounds. Key studies highlight its effectiveness in achieving high fine particle fractions, rapid dissolution, and superior aerosol performance, as demonstrated in formulations for cystic fibrosis treatment (Yu, 2021; Shahin, 2021) and inhalable antifungals (Liao, 2019). Additionally, vaccines produced via SFD have showed remarkable results in immune responses in the respiratory tract, making this method appealing in the preparation of immunotherapy treatments (Saluja, 2010; Amorij, 2007). SFD products’ high porosity and amorphous particle state have also improved bioavailability in small-molecule drugs (Lucas, 2022). These findings demonstrate the potential of this method to enhance therapeutic effectiveness, stability, and targeted delivery across a range of applications.”
In addition, we also provided several sentences in the Pharmaceutical Applications Section (for example: l. 382, 387, 397).

Reviewer 2 Report
Comments and Suggestions for Authors
The paper is a new review on Spray-Freeze-Drying; comparing to a previous one by other authors consider also food and ceramic/catalytic materials (and thus it addresses a larger audience than that of Pharmaceutics) and make also a comparison with traditional freeze-drying and spray-drying.
It is a work of reasonable quality, with a sufficient level of analysis and discussion, well written.
It can be published, after some revision. The list of points to be addressed is listed below.
1) Line 100: SD starts with residual moisture of 10%: this is a reasonable value, but actually residual moisture at end of PD depends on the nature of excipients, and can be higher.
2) Line 168 (and Figure 5): images are taken from previous review paper [20], quoted, but in the text reference is to paper [19]
3) In table 1 “Ultrasonic nozzle” is not a parameter, and it is not clear its effect on particle size distribution; this effect should be better explained, and the line reformulated (or omitted and discussed in the text)
4) In Table 1, “Duration of primary drying” is confusing as a parameter, as it actually is a function of porosity and particle size. If the authors refer to shrinkage effect, prolonging unnecessarily primary drying, it may be, but should not be put there.
5) Line 197-8: compare ref given with those in Table 1
6) Line 278-280: what the authors mean by “increased fluidization speed”? This part should be better explained, is quite confusing. Initially they discuss of freezing time (a typo? Or explain what are referring to), then relate to gas velocity? But the controlling resistance should be on the particle side.
7) Section 6: please note that the initial paragraph is duplicated from a previous section
8) Line 496: explain “the nanoparticles are amorphously embedded in the excipients” and previous reference to “particles”. This part is confused, unfolding of protein is correctly discussed, but reference to particles is unclear. If the authors want to discuss behaviour of suspensions, should be done in better detail.
9) Line 578: wrong numeration (section 7?)
10) Line 599: I disagree with the authors, about the fact that FD require large amounts of dry cold gas, it actually requires no cold gas; this is true for fluidized bed atmospheric freeze drying, but here authors are referring to standard FD, it seems.
11) Verify text: line 168, sand? Line 215, teds?
12) Bibliography is very very poor; apart the style that is variable from record to record, in many cases just the first author et al., is given, in many cases record is incomplete (paper number or pages missing, or just the title is given).
Author Response
Reply to Reviewer 2
We would like to thank the Reviewer for the useful comments and suggestions made on our manuscript, which helped us to improve it substantially. Please find below the changes made on our manuscript following the Reviewer’s recommendations.
The paper is a new review on Spray-Freeze-Drying; comparing to a previous one by other authors consider also food and ceramic/catalytic materials (and thus it addresses a larger audience than that of Pharmaceutics) and make also a comparison with traditional freeze-drying and spray-drying. It is a work of reasonable quality, with a sufficient level of analysis and discussion, well written. It can be published, after some revision. The list of points to be addressed is listed below.
- Line 100: SD starts with residual moisture of 10%: this is a reasonable value, but actually residual moisture at end of PD depends on the nature of excipients, and can be higher.
Thank you for this useful observation. We removed the percentage of residual moisture from the text so that misconceptions are avoided.
- Line 168 (and Figure 5): images are taken from previous review paper [20], quoted, but in the text reference is to paper [19]
We modified the references.
- In table 1 “Ultrasonic nozzle” is not a parameter, and it is not clear its effect on particle size distribution; this effect should be better explained, and the line reformulated (or omitted and discussed in the text)
Thank you for this useful comment. The authors proceeded with the modification of the manuscript according to this note. Regarding the table, actually, the ultrasonic nozzle is not a parameter, however in this table we compare the use of the ultrasonic nozzle to the use of a 2-Fluid nozzle, and a narrower particle size distribution is observed in the first case. However, if the Reviewer believes that this parameter should be removed from the table, we are in agreement. Regarding the better explanation, we modified the text accordingly: Another type of nozzle is the ultrasonic nozzle. It achieves the desired droplet characteristics through vibration, with the critical parameter being the frequency of vibration. Lower frequency causes an increase in the droplet size as the standing sinusoidal longitudinal wave responsible for the atomization has less energy. The droplet size distribution is narrower compared to the 2-fluid nozzle due to better control over the droplet formation. The 2-fluid nozzle requires a high atomizing gas flow rate and thus small sample sizes cannot be as easily handled (Schiffter et al., 2010).
- In Table 1, “Duration of primary drying” is confusing as a parameter, as it actually is a function of porosity and particle size. If the authors refer to shrinkage effect, prolonging unnecessarily primary drying, it may be, but should not be put there.
The parameter of “duration of primary drying” was excluded from the manuscript so that this part won’t be confusing for the readers.
- Line 197-8: compare ref given with those in Table 1
The references were modified accordingly.
- Line 278-280: what the authors mean by “increased fluidization speed”? This part should be better explained, is quite confusing. Initially they discuss of freezing time (a typo? Or explain what are referring to), then relate to gas velocity? But the controlling resistance should be on the particle side.
The comment of the Reviewer is totally right. The text was modified: “For fluidized bed drying, a critical parameter regarding drying time is the gas flow rate, as a larger quantity of air allows for more ice to be sublimated (Mumenthaler & Leuenberger, n.d.). Decreasing the pressure results in higher drying speeds, as the lower partial pressure of the water vapour in the drying air creates an increased mass transfer force. However, secondary drying is not possible, as the electrostatic forces cause particle adhesion to the walls (Leuenberger et al., 2006)”.
- Section 6: please note that the initial paragraph is duplicated from a previous section
The initial paragraph is excluded.
8) Line 496: explain “the nanoparticles are amorphously embedded in the excipients” and previous reference to “particles”. This part is confused, unfolding of protein is correctly discussed, but reference to particles is unclear. If the authors want to discuss behaviour of suspensions, should be done in better detail.
The authors totally agree with this comment and therefore, proceeded to the following modification so that potential misinterpretations are prevented for the readers. The nanoparticles refer to the drug solution, that can be amorphously embedded in the excipients and be distributed into them while processing. In the revised text the ‘nanoparticles’ have been replaced with ‘drug solution’. (l. 577)
9) Line 578: wrong numeration (section 7?)
The numeration is corrected.
10) Line 599: I disagree with the authors, about the fact that FD require large amounts of dry cold gas, it actually requires no cold gas; this is true for fluidized bed atmospheric freeze drying, but here authors are referring to standard FD, it seems.
We would like to thank the Reviewer for this useful correction. The text should have specified that this applies to Atmospheric SFD. The revised version is found” This is primarily due to the low processing temperatures, the requirement for vacuum conditions or large amounts of dry cold gas in the case of atmospheric SFD, all of which jeopardize the economic viability of the SFD operation”.
11) Verify text: line 168, sand? Line 215, teds?
The typos are corrected.
12) Bibliography is very very poor; apart the style that is variable from record to record, in many cases just the first author et al., is given, in many cases record is incomplete (paper number or pages missing, or just the title is given).
The style was corrected so that it does not vary and it is according to the guidelines of the journal. The bibliography can be edited in the end of the review process during the proofs.

Reviewer 3 Report
Comments and Suggestions for Authors
Overall, the work presented by Sartzi et al is interesting and relatively novel as it brings up different aspects of spray freeze process compared with othe rtehcnologies. However tehre are few comments that need revision befor epublication:
1. I think the paper is very theoretical, whihc is important but also I woudl like to see more real examples with the final particle characteristics.
2. In the tables, I am missing key information, whihc is the particle size, the drug loading and the yield of the process? Otherwise in the curren tstate it is just a list of potential application but I am not sure whihc is the advantage fo using this technology over conventiona ones, how much yiled can I get or how much more I can control particle size over conventional sprya dyring?
This qesutions shoudl be addressed in the review.
3. How many products that are in the market are using this tehcnology?
4. A section fo future persperctives woudl be ideal to understand the potential of this tehcnology in the future.
5. Regarding the pulmonayr drug deliveyr field, it is critical to understand the differences obtain between SF adn SD for similar polymers or carbohydrates. Beign able to control particle size is key to ensure optimal lung deposition, so for example, using polymers or carbohydrates how this cna be improved? See paper: "Heparin-azithromycin microparticles show anti-inflammatory effects and inhibit SARS-CoV-2 and bacterial pathogens associated to lung infections" or "Targeting lung macrophages for fungal and parasitic pulmonary infections with innovative amphotericin B dry powder inhalers"
6. Also, a key point is how much water is left over in the particles after the process as this is cirticla for physical stabitliy and microbiological issues?
7. Further discussion about implementation in industrial settings is key.
Author Response
Reply to Reviewer 3
We would like to thank the Reviewer for the useful comments and suggestions made on our manuscript, which helped us to improve it substantially. Please find below the changes made on our manuscript following the Reviewer’s recommendations.
Overall, the work presented by Sartzi et al is interesting and relatively novel as it brings up different aspects of spray freeze process compared with othe rtehcnologies. However tehre are few comments that need revision befor epublication:
- I think the paper is very theoretical, whihc is important but also I woudl like to see more real examples with the final particle characteristics.
More details about the characteristics of the final SFD products, such as fine particle fraction, particle diameter etc. are provided in Table 2,3 and 4 (Section 5: Application of SFD) so that the applications are enriched and the paper becomes less theoretical.
- In the tables, I am missing key information, whihc is the particle size, the drug loading and the yield of the process? Otherwise in the curren tstate it is just a list of potential application but I am not sure whihc is the advantage fo using this technology over conventiona ones, how much yiled can I get or how much more I can control particle size over conventional sprya dyring?
We would like to thank the Reviewer for this helpful comment. A column was added in Table 2 and 3, that includes information on particle size, drug loading and yield, on articles where such information is provided.
This qesutions shoudl be addressed in the review.
- How many products that are in the market are using this tehcnology?
The authors totally agree with the Reviewer that this information will be very useful and interesting for the readers. However, after an extensive research that was performed for many days in different sources, such as latest books, LinkedIn pages, review papers, market reports, this information could not be found. If the Reviewer could suggest a source, it would be more than welcome and it could be added in the manuscript.
- A section fo future persperctives woudl be ideal to understand the potential of this tehcnology in the future.
The Section 8. “Future Needs” discusses the future perspectives and requirements of this technology and it is enriched in the new version. Please inform us if we should proceed to further additions.
- Regarding the pulmonary drug delivery field, it is critical to understand the differences obtain between SF adn SD for similar polymers or carbohydrates. Beign able to control particle size is key to ensure optimal lung deposition, so for example, using polymers or carbohydrates how this cna be improved? See paper: "Heparin-azithromycin microparticles show anti-inflammatory effects and inhibit SARS-CoV-2 and bacterial pathogens associated to lung infections" or "Targeting lung macrophages for fungal and parasitic pulmonary infections with innovative amphotericin B dry powder inhalers"
Several paragrpahs are added in the manuscript to improve the understanding of the differences among the SF, SD and SFD: In pulmonary drug delivery particle size, morphology and density are of major significance for optimizing lung deposition and therapeutic efficacy (Chaurasiya & Zhao, 2020). As mentioned previously, SD produces smaller, denser particles, that are appropriate for fine-tuning aerodynamic properties (Hoppentocht et al., 2014). Polymers like poly(lactic-co-glycolic acid) (PLGA) and carbohydrates, such as mannitol and lactose, are commonly used in SD formulations to enhance stability and control drug release (Pilcer & Amighi, 2010). In contrast, SFD produces larger particles, with high porosity, reduced density and a spherical structure, enhancing dispersibility and lung deposition due to favorable aerodynamic characteristics. These can also reduce particles aggregation and enhance lung distribution, making it an appealing technique to produce powders that target specific pulmonary regions (Chhabra et al., 2024).
Employing SFD or SD to alter particle size and drug release rates of drug formulations can have direct impact on their efficacy. For example, Ananya et al. (2025) have demonstrated how effective polymer-based solutions, with anti-inflammatory properties, are in pulmonary drug delivery. The study proposes the combination of heparin and azithromycin to reduce inflammation and inhibit pathogens such as SARS-CoV-2 and bacteria associated with lung infections (Anaya et al., 2025). De Pablo et al. (2023) explored the use of SD to develop dry powder formulations that can target lung macrophages for treating fungal and peracetic infections. The authors highlighted how particle engineering can help optimize drug delivery to the deep lung, while enhancing macrophage uptake (de Pablo et al., 2023). The use of innovative excipients may further improved lung deposition by controlling particle size and density though the atomization process. By carefully selecting polymers or carbohydrates in SD and SFD, controlling the particle size and density, the enhancement of drug targeting within the lung and the improvement of therapeutic efficacy can be achieved.
- Also, a key point is how much water is left over in the particles after the process as this is cirticla for physical stabitliy and microbiological issues?
We would like to thank the Reviewer for this important comment. Information on the moisture content of the produced products is included in Table 2 and 3, according to the availability of such information in the research. Furthermore, the following paragraph discussing the importance of the residual moisture was added to section 6.1 “SFD has been proven to be effective for a multitude of applications as seen in Table 2 and 3. In a study on docosahexaenoic acid encapsulation it was found that the resulting product had comparable residual moisture levels to the other conventional drying methods. Specifically, SFD powder had 3.66% FD 5.49% and SD 2.40%. The poorer result in comparison to SD, is suspected to be the low temperatures used for SFD (Karthik & Anandharamakrishnan, 2013). Great importance in drying methods, is given to the residual moisture content of the products(Ishwarya et al., 2015). A high enough water content not only affects the physical characteristics of the produced particles, as water decreases the Tg resulting in a less stable solid structure (Tonnis et al., 2014), but it may also affect the stability of the active ingredients such as an antigen (Maa et al., 2004). Furthermore, in food applications, moisture content is a crucial parameter, determining factors such as shelf life as it affects mold growth and agglomeration (Elik et al., 2021). The reduction of the water content below a certain threshold is critical for ensuring product acceptability (Cao et al., 2020). “
- Further discussion about implementation in industrial settings is key.
The implementation in the industrial scale is analytically described in the last 2 paragraphs of Section 8. “Future Needs” and of course throughout the whole text in the Section of the Applications. Please inform us if you would like us to proceed to further discussion.

Round 2
Reviewer 2 Report
Comments and Suggestions for Authors
Points raised have been addressed, and manuscript improved with some additions.
References have been added, and reference style moved to Harvard.
The authors declared: "The style was corrected so that it does not vary and it is according to the guidelines of the journal. The bibliography can be edited in the end of the review process during the proofs."
Actually some variation of style are still present, and some data missing. Personally I disagree with the approach of the authors, but leave the editor to manage.
Reviewer 3 Report
Comments and Suggestions for Authors
Authros have addressed all the comments and manuscrito si ready for publication